# The Prevalence, Seroprevalence, and Risk Factors of Tick-Borne Encephalitis Virus in Dogs in Lithuania, a Highly Endemic State

**DOI:** 10.3390/v15112265

**Published:** 2023-11-17

**Authors:** Evelina Simkute, Arnoldas Pautienius, Juozas Grigas, Paulina Urbute, Arunas Stankevicius

**Affiliations:** 1Laboratory of Immunology, Department of Anatomy and Physiology, Lithuanian University of Health Sciences, Tilzes Str. 18, LT-47181 Kaunas, Lithuania; arnoldas.pautienius@lsmu.lt (A.P.); juozas.grigas@lsmu.lt (J.G.); paulinaurbute@gmail.com (P.U.); arunas.stankevicius@lsmu.lt (A.S.); 2Institute of Microbiology and Virology, Lithuanian University of Health Sciences, LT-47181 Kaunas, Lithuania

**Keywords:** tick-borne encephalitis, TBEV prevalence, TBEV seroprevalence, TBEV in dogs, RT-nPCR

## Abstract

The rising awareness and increasing number of case reports of tick-borne encephalitis (TBE) in dogs indicate that the virus might be an important tick-borne pathogen in dogs, especially in endemic areas. Therefore, the aim of the present study was to investigate the prevalence rate of TBEV RNA and TBEV-specific antibodies in clinical samples of dogs living in a highly endemic region of Lithuania and to evaluate the main risk factors for severe disease course and death. The blood samples (*n* = 473) of dogs were collected in two veterinary clinics in central Lithuania. Tick-borne encephalitis virus (TBEV) RNA was detected in 18.6% (88/473; CI 95% 15.2–22.4) and TBEV-specific antibodies were found in 21.6% (102/473; CI 95% 17.9–25.6) of dog blood serum samples after confirmation with a virus neutralization test. The death/euthanasia rate was 18.2% (16/88; CI 95% 10.8–27.8) in PCR-positive dogs. Male dogs were more likely to develop neurological symptoms (*p* = 0.008). Older dogs (*p* = 0.003), dogs with the presence of neurological symptoms (*p* = 0.003), and dogs with the presence of TBEV-specific antibodies (*p* = 0.024) were more likely to experience worse outcomes of the disease. The results of the present study demonstrate that TBEV is a common and clinically important pathogen in dogs in such endemic countries as Lithuania.

## 1. Introduction

Tick-borne encephalitis virus (TBEV) is one of the most important tick-borne viruses in human medicine. The virus can cause neurological symptoms with long-term health impairment and can be fatal [1]. TBEV belongs to the family *Flaviviridae* and is endemic in Europe and Northeastern Asia [2]. For the past decade, Lithuania has recorded one of the highest prevalence rates of human TBE cases in Europe [3]. Although many species of animals can be infected by TBEV, clinical symptoms are mainly observed in humans and less commonly in horses and dogs [4].

In endemic areas, TBE is considered one of the most important infections of the central nervous system (CNS) in dogs [5]. In some endemic locations of Europe, the seroprevalence of TBEV-specific antibodies in dogs has been reported to be 30–40% [6,7,8]. Studies suggest that the majority of TBE cases in dogs are asymptomatic, although cases of peracute, acute, and chronic courses of the disease were reported [9,10,11]. The clinical symptoms of TBE in dogs are similar to the symptoms reported in human patients and a biphasic course of the disease is prominent in dogs [5,12]. At the first stage, fever, apathy, anorexia, and gastrointestinal symptoms are mainly reported both in human patients and in dogs. If the second stage of TBE develops, neurological symptoms are mainly observed [5,13]. Although neurological symptoms in dogs living in TBEV-endemic regions strongly imply TBE as a differential diagnosis, studies show that TBE is underdiagnosed in dogs [11,14,15].

Currently, the diagnosis of TBE mainly relies on the detection of specific IgM antibodies in cerebrospinal fluid (CSF) and/or in blood serum samples [3]. Although cross-reactions with other flaviviruses are conceivable, an increase in IgG antibodies evaluated with an ELISA in recollected samples might aid in the diagnosis of TBE [16]. For an early differential diagnosis of TBE, viral RNA detection using PCR might be a valuable tool [3]. In addition, the presence of neurological clinical symptoms, anamnesis of tick-bite, and living or visiting TBE endemic areas may all point to TBE [17].

The present study sought to investigate the prevalence of TBEV RNA and TBEV-specific antibodies in dogs living in Lithuania for the first time, as well as to evaluate the risk factors for severe disease course and mortality.

## 2. Materials and Methods

### 2.1. Sample Collection and Data of Anamnesis

Two veterinary clinics located in Kaunas (central part of Lithuania) agreed to participate in the present study. Prior to conducting the investigation, explicit consent was obtained from the owners of the dogs, granting authorization for the utilization of clinical samples. Randomly collected blood samples from dogs (*n* = 473) with various diseases were obtained from clinics between May 2020 and December 2021. Additionally, 3 samples of cerebrospinal fluid (CSF) were collected and analyzed. All samples were stored at −80 °C for further analysis. Data on clinical anamnesis and outcome of disease were collected retrospectively starting one month before sample collection and followed up to six months thereafter. Dogs of 45 different breeds were included in the study and the most common anamnesis was apathy, inappetence, and gastrointestinal symptoms (Appendix A). Concurrent tick-borne diseases were diagnosed at veterinary clinics using blood smears, focusing on the detection of *Babesia* spp. and *Anaplasma phagocytophilum*. Anti-*A. phagocytophilum* antibodies were detected by chromatographic immunoassay test (Anigen Rapid CaniV-4 Test Kit (BioNote, INC. Hwaseong, Republic of Korea). The attached ticks (*n* = 117) were collected when they were found on the bodies of dogs at the same visit at the veterinary clinic when blood serum samples were collected. Each tick was individually treated with liquid nitrogen, homogenized using mortar and pestle, mixed with 1000 μL of Modified Eagle’s Medium (MEM, Gibco, Waltham, MA, USA), and stored at −80 °C for further analysis.

### 2.2. Sample Analysis with ELISA, VNT, and RT-nPCR

The serum samples were analyzed using commercially available quantitative enzyme-linked immunosorbent assay (ELISA) (VetLine TBE/FSME ELISA, NovaTec Immundiagnostica GmbH, Dietzenbach, Germany) according to the manufacturer’s instructions. To confirm positive ELISA results, a virus neutralization test was performed according to the instructions described before [18].

Total RNA was extracted from blood serum (*n* = 473) and CSF (*n* = 3) samples using GeneJET RNA Purification Kit (Thermo Scientific, Waltham, MA, USA) according to the manufacturer’s instructions. Serum and whole blood samples of dogs and suspensions of ticks and isolates in cell cultures were analyzed with reverse transcription nested polymerase chain reaction assay (RT-nPCR) using DreamTaq Green PCR Master Mix and primers as previously described [19].

### 2.3. Virus Isolation

The suspensions of ticks, blood serum, and CSF samples were centrifuged at 12.000 g for 5 min and filtered with a 0.22 μm pore size microfilter (Techno Plastic Products AG, Trasadingen, Switzerland). Cell cultures of Vero (ATCC No. CCL-81) and Marc-145 (ATCC No. CRL-12231) were seeded in a maintenance medium containing Modified Eagle’s Medium (MEM, Gibco, USA) supplemented with 10% heat-inactivated fetal bovine serum (FBS; Gibco, USA), 100 U/mL penicillin, and 100 μg/mL streptomycin. Dulbecco’s Modified Eagle’s Medium (DMEM, Gibco, USA) was used for murine neuroblastoma cells (Neuro-2a ATCC No. CCL-131) instead of MEM. All cells were incubated at 37 °C in 5% CO_2_ overnight. Cells were inoculated with prepared tick suspensions or clinical samples from the dogs the following day. Cell cultures were examined for the occurrence of cytopathic effects through three serial passages which were performed in triplicates including triplicates of positive and negative controls for each round of analysis. After each serial passage, cell suspensions were harvested for RNA extraction. Successful virus isolation in cell cultures was confirmed using RT-nPCR. Partial genome sequencing targeting the NCR region of TBEV was used for internal confirmation of the specificity of detected TBEV RNA.

### 2.4. Statistical Analysis

The confidence intervals were calculated using the exact binomial method. To determine the significance of the risk variables, binary logistic regression analysis and the chi-square test were used. Standardized residuals were employed as post-hoc analysis after statistically significant chi-square tests. The statistical analysis and visualizations were performed utilizing the programming language R 4.3.2.

## 3. Results

### 3.1. Prevalence of TBEV and TBEV-Specific Antibodies in Dogs

A total number of 473 dog blood samples were analyzed using RT-nPCR and ELISA in the present study. Viral RNA was detected in 18.6% (88/473; CI 95% 15.2–22.4) of dog blood/serum samples. In addition, viral RNA was detected in two out of three CSF samples. TBEV-specific antibodies were detected in 21.6% (102/473; CI 95% 17.9–25.6) of dog blood serum samples after confirmation with a virus neutralization test (Figure 1A). No statistically significant associations were found between the antibody and TBEV RNA prevalence rates in dog blood serum samples based on sample collection time (Figure 1B).

The median age of PCR-positive dogs was 7 years, ranging from 2 months to 15 years. The rate of death/euthanasia was 18.2% (16/88; CI 95% 10.8–27.8) in PCR-positive dogs. Because of severe neurological symptoms or poor prognosis, 12.5% (11/88; CI 95% 6.4–21.3) of PCR-positive dogs were euthanized while the remaining 5.7% (5/88; CI 95% 1.9–12.8) died regardless of intense care and symptomatic treatment. Tick bites were reported by owners in 29.6% (26/88; CI 95% 20.3–40.2) of PCR-positive dogs over a two-week period. Other tick-borne diseases were concurrently diagnosed in 19.3% (17/88; CI 95% 11.7–29.1) of PCR-positive dogs (*Babesia* spp. and *A. phagocytophilum* were diagnosed in 17 and 3 dogs, respectively). In addition, dogs of 36 different breeds were found to be PCR-positive, including small dog breeds.

TBEV-specific antibodies were detected in 26.1% (23/88; CI 95% 17.3–36.6) of PCR-positive dogs with a mean antibody level of 189 NTU/mL (range 29.7–525 NTU/mL). The median age of seropositive viremic dogs was 8 years, ranging from 1 to 14 years. Dogs older than 1 year of age were found to be seropositive and viremic statistically significantly more often than dogs under the age of 1 year (37.5% (95% CI 26.3–49.7) and 0%, respectively, (*p* = 0.003)). In addition, 16.7% (79/473; CI 95% 13.5–20.4) of the dogs in the study were seropositive but PCR-negative.

### 3.2. Risk Factors for Clinical TBE Infection in Dogs and Lethal Outcomes

Statistical analysis revealed that male dogs (43.7% (95% CI 29.5–58.8)) were more likely (*p* = 0.008) to develop neurological symptoms compared to females (17.5% (95% CI 7.3–33.0)). The lethal outcome/euthanasia of PCR-positive dogs with neurological symptoms was associated with older age. The mean age of recovered and deceased/euthanized dogs was 5.6 and 11.8 years, respectively (*p* = 0.003). Other causes of death/euthanasia, such as neoplastic processes, trauma, or hydrothorax were not related to the age of PCR-positive dogs. Moreover, dogs with neurological symptoms had a higher risk of experiencing a worse outcome (*χ*^2^; = 11.57, *df* = 2, *p* = 0.003) resulting in a significantly lower number of recovered dogs with neurological symptoms (27.8% (95% CI 19.3–36.6)) compared to the number of recovered dogs without neurological symptoms (72.2% (95% CI 63.7–80.7)).

The chi-square test of independence revealed a statistically significant association between the antibody status and the outcome of the disease (*χ*^2^; = 5.05, *df* = 1, *p* = 0.024). Subsequent post-hoc analysis revealed that the frequencies of recovered dogs were lower by 14.2% in the seropositive group and higher by 8.6% in the seronegative group as compared to the predicted model findings. In contrast, the incidences of death/euthanasia were 65.5% higher in seropositive dogs and 39.7% lower in seronegative dogs. The results imply that the presence of TBEV-specific antibodies in the blood serum of PCR-positive dogs might be related to a higher risk of death/euthanasia.

### 3.3. TBEV Seroprevalence and Associations with Clinical Symptoms and the Presence of other Tick-Borne Diseases

Out of the 102 seropositive dog blood serum samples confirmed with a virus neutralization test, 13.7% (14/102; CI 95% 7.7–22) were considered borderline (20–30 NTU/mL) with an ELISA test, and 86.3% (88/102; CI 95% 78–92.3) were positive (>30 NTU/mL) (Figure 2A–C). Statistical analysis revealed significantly higher antibody levels in dogs with potentially TBE-related symptoms compared to dogs with non-specific symptoms (*p* = 0.002) and dogs with non-TBE-related symptoms (*p* = 0.001) (Figure 2D).

Other tick-borne diseases were diagnosed in 14.7% (15/102; CI 95% 8.5–23.1) of dogs with TBE-specific antibodies: protozoa of *Babesia* spp. were found in blood samples of 13 dogs and 2 dogs had specific antibodies against *A. phagocytophilum*. Moreover, dogs diagnosed with other tick-borne diseases had significantly (*p* = 0.046) higher levels of TBEV-specific antibodies (Figure 2E).

Clinical neurological symptoms potentially related to TBE were found in 13.7% (14/102; CI 95% 7.7–22) of seropositive dogs. Non-specific symptoms such as hyperthermia, apathy, and gastrointestinal symptoms (vomiting and/or diarrhea) were observed in 32.4% (33/102; CI 95% 23.4–42.3) of seropositive dogs. Other diseases were diagnosed in 39.2% (40/102; CI 95% 29.7–49.4) of seropositive dogs with various symptoms that were not considered to be associated with TBE (skin allergies, pyometra, kidney or liver failure, trauma, oncologic processes, and other diseases).

On physical examination, neurological symptoms were observed in 31.8% (28/88; CI 95% 22.3–42.6) of PCR-positive dogs. TBEV-specific antibodies were detected in 32.1% (9/28; CI 95% 15.9–52.4) of PCR-positive dogs with neurological symptoms. The dogs with neurological symptoms had a median age of 8 years, ranging from 4 months to 15 years. Neurological symptoms were considered moderate or severe in 12.5% (11/88; CI 95% 6.4–21.3) of viremic dogs. Episodes of seizures, tetraparesis, impaired vision, deficits of facial reflexes, altered mental status, behavior changes, and cognitive function impairment were observed. Five of six dogs with seizures died, four of which did not show hyperthermia, thus implying that the absence of fever in the presence of severe neurological symptoms might be related to a worse outcome of TBE. Mild neurological symptoms were present in 19.3% (17/88; CI 95% 11.7–29.1) of PCR-positive dogs. Difficulties standing up, reluctance to walk, presence of a stiff walk, back hyperesthesia (without organic defects in the musculoskeletal system), and impaired coordination were observed.

In 68.2% (60/88; CI 95% 57.4–77.7) of PCR-positive dogs, no TBE-related neurological symptoms were noticed. Various concurrent diseases were diagnosed in 25% (22/88; CI 95% 16.4–35.4) of dogs. Gastrointestinal symptoms (vomiting/diarrhea) were present in 37.5% (33/88; CI 95% 27.4–48.5) of PCR-positive dogs (of them, 17.1% (15/88; CI 95% 9.9–26.6) of dogs also had neurological symptoms). Other tick-borne diseases were diagnosed in 13.6% (12/88; CI 95% 7.3–22.6) of dogs. Non-specific symptoms such as apathy and loss of appetite were observed in 11.4% (10/88; CI 95% 5.6–19.9) of PCR-positive dogs. Hyperthermia was present in 37.5% (33/88; CI 95% 27.4–48.5) of PCR-positive dogs: 15.9% (14/88; CI 95% 9–25.3) were diagnosed with other concurrent tick-borne diseases; 13.6% (12/88; CI 95% 7.3–22.6) had TBE-related symptoms; and the remaining 10.2% (9/88; CI 95% 4.8–18.5) of dogs had other possible reasons for fever (trauma, neoplastic processes, and other).

Neurological symptoms were present in 12.7% (10/79; CI 95% 6.2–22.1) of seropositive and PCR-negative dogs. Behavior changes were present in two dogs: One dog was found by its owners in a hypersensitive state and was hyperactive, disoriented, and injured himself. The episodes of hypersensitivity lasted for a few days and an episode of impaired vision was present. Another dog was hyperactive and anxious and had episodes of extremely increased appetite for eating grass and licking carpets. Both dogs had high antibody levels (473 NTU/mL and 494 NTU/mL, respectively) and both dogs recovered. Mild neurological symptoms were present in 8.9% (7/79; CI 95% 3.6–17.4) of seropositive dogs and one dog showed uncoordinated movements and episodes of falling down.

No potentially TBE-related symptoms were noticed in 48.1% (38/79; CI 95% 36.7–59.6) of seropositive and PCR-negative dogs. Non-specific symptoms (apathy and loss of appetite) were present in 29.1% (23/79; CI 95% 19.4–40.4) of dogs without final diagnosis and gastrointestinal symptoms were present in 26.6% (21/79; CI 95% 17.3–37.7) of seropositive and PCR-negative dogs.

### 3.4. TBEV RNA Prevalence in Ticks, Collected from Dogs

Attached *Ixodes ricinus* (*n* = 112) and *Dermacentor reticulatus* (*n* = 5) ticks were collected from the dogs and tick suspensions were analyzed using RT-nPCR. Concurrently, tick suspensions were inoculated on Neuro-2a, Vero, and Marc-145 cells. Overall, TBEV RNA was detected in 34.2% (40/117; CI 95% 25.7–43.5) of tick suspension samples (33.9% (38/112; CI 95% 25.8–43.1) in *Ixodes ricinus* and 40.0% (2/5; CI 95% 11.7–76.9) in *Dermacentor reticulatus*). The number of positive samples increased significantly (*p* = 0.001) to 56.4% (66/117; CI 95% 46.9–65.6) in Marc-145, 58.1% (68/117; CI 95% 48.6–67.2) in Vero, and 60.7% (71/117; CI 95% 51.2–69.6) in Neuro-2a cells after three serial passages (Figure 3A,B). However, no statistically significant relations were found between the viral RNA present in the dog blood samples and the ticks, collected from the respective dogs.

### 3.5. Virus Isolation from PCR-Positive Dogs’ Serum and CSF Samples

Due to a short viremia and rapid viral clearance, successful attempts to isolate TBEV in cell cultures from clinical dog samples are rare. However, we successfully isolated TBEV from the blood serum (*n* = 7) and CSF (*n* = 1) samples of PCR-positive dogs in Neuro-2a, Vero, and Marc-145 cell cultures. Successful virus isolation was confirmed using RT-nPCR after the second or third passage in the cells.

## 4. Discussion

The diagnosis of TBE currently relies mainly on detecting TBEV-specific antibodies in CSF or in blood serum samples [3]. However, serological tests may be negative in the early phases of TBE, as was demonstrated in the present study with only 26.1% (23/88; CI 95% 17.3–36.6) of PCR-positive dogs being seropositive for TBEV. In addition, cross-reactions with IgG antibodies of other flaviviruses are possible [16]. The detection of TBEV RNA in 18.6% (88/473) of the randomly collected dog blood samples is in accordance with the results of other authors. A similar study conducted in the Czech Republic reported a prevalence rate of 12.6% (20/159) of PCR-positive dogs in the TBE-endemic region [20]. The high TBEV RNA prevalence in dogs might be related to reduced or delayed immune response due to various diseases such as babesiosis, neoplasia, or chronic diseases. Also, treatment with corticosteroids might be related to longer-lasting viremia [21]. In addition, the age, breed, physiological and immune status of the dog, the specific TBEV strain, or infectious dose may also play an important role in predisposing clinical TBE [22]. Although TBEV RNA detection using PCR is not the preferred method to confirm TBE infection in humans or dogs, Saksida et al. [23] concluded that RT-PCR is an efficient method for an early diagnosis of TBE while specific antibodies are still undetectable. Therefore, the findings suggest that viral RNA detection using PCR might be a valuable tool in diagnosing TBE in dogs in endemic regions, especially in the early or acute stages of the disease.

TBEV-specific antibodies were detected in 21.6% (102/473) of the randomly collected dog blood samples and in 32.1% (9/28) of PCR-positive dogs with neurological symptoms. Although the results are consistent with reports from other European countries, a substantially greater seroprevalence of 37.5% (113/301) was found in horses in a prior investigation in Lithuania [24]. In healthy dogs, the highest reported prevalence rate of TBE-specific antibodies detected using ELISA was 30.4% (17/54) in endemic regions of Germany [8], 30.4% (38/125) in Denmark [7], and 40% (8/20) in the Åland Islands of Finland [6]. However, VNT confirmed the ELISA results only in the Danish island investigation. Following the viral neutralization test, the seroprevalence rate was 4.8% (6/125) in only one area in Bornholm, a Danish island in the Baltic Sea [7], implying that the true seroprevalence might be lower in other countries as well. The reported prevalence rate of TBEV-specific antibodies in dogs with neurological symptoms varied between 11.3% (18/159) in the blood serum samples of dogs from an endemic region in the Czech Republic [20]; 20.2% (110/545) in Austria [25], and 53.6% (30/56) in Germany (30.4% (17/56) also had detectable antibodies in CSF [8]. The results of high seroprevalence show that TBEV in dogs in Lithuania is a common tick-borne pathogen, similar to other TBEV-endemic areas.

A death/euthanasia rate of 18.2% (16/88; CI 95% 10.8-27.8) is much lower than the 33.3% (18/54) reported by Kleeb et al. [5]. We, on the other hand, analyzed randomly collected samples of dogs, while Kleeb et al. [5] conducted a retrospective study of confirmed TBE cases in dogs with clinical symptoms. In the present study, all PCR-positive dogs under the age of 1 year were TBEV seronegative. The most likely explanation is that these dogs were infected with TBEV for the first time in their life and antibodies were not detectable at the time of testing. A study conducted by Salat et al. [26] found a statistically significant correlation between age and VNT seropositivity. However, the data on TBEV seroprevalence in dogs under the age of 1 year is scarce because older dogs are mainly included in the studies since they are more likely to have had previous contact with ticks and, therefore, tick-borne pathogens [7,26]. In addition, dogs of small breeds (less than 15 kg weight) were omitted from some of the TBE seroprevalence studies for the same reason as young dogs [7]. However, the results of our study show that differential diagnosis of TBE in endemic areas should not be ruled out due to the young age or small size of the dog.

We looked at the putative risk factors of PCR-positive dogs that might be associated with the severity of clinical neurological symptoms and worse outcomes of the disease. Statistical analysis revealed that male dogs were prone to developing neurological symptoms. Older dogs and dogs with neurological symptoms were more likely to experience worse outcomes, compared to younger dogs and dogs without neurological symptoms. Furthermore, PCR-positive and seropositive dogs (mainly with severe clinical symptoms) were more likely to experience worse outcomes, possibly because of a delayed antibody response in the acute monophasic course of infection when neurological symptoms develop in the presence of viremia. In accordance with our findings, Kleeb et al. [5] reported that older dogs and dogs with seizures had an increased risk of death. Studies in human patients concluded that older age and male gender were associated with a more severe course of the disease [27,28]. In addition, the delayed immune response of TBEV-Ig antibodies and the monophasic course of the disease were the predictor factors of the severe form of TBE infection in human patients [29,30].

Significantly higher antibody levels were detected in the serum samples of dogs with possibly TBE-related symptoms compared with dogs without TBE-related symptoms. Most likely, these dogs could have had a recent TBE infection which was asymptomatic or not suspected and diagnosed. TBE is thought to be underdiagnosed in dogs, owing to mild or non-specific clinical symptoms or unawareness of the possible disease in severe cases [9,11]. Coinfection with other tick-borne pathogens was diagnosed in 19.3% (17/88; CI 95% 11.7–29.1) of TBEV PCR-positive dogs. In addition, dogs diagnosed with other concurrent tick-borne diseases were more likely to have TBEV-specific antibodies and the levels of antibodies were higher. These findings suggest that dogs with a clinical history of several tick-borne diseases are possibly living in or visiting areas with a high risk of tick-bite and tick-borne infections and therefore could be used as indicators to evaluate the risk of tick-borne zoonoses in humans. It is known that dogs are 50–100 times more likely than human beings to come into contact with TBEV-infected ticks [9]. Furthermore, dogs are known sentinels for TBE foci detection and were used as TBE indicators in countries previously considered to be TBE-free [31].

To the best of our knowledge, we are the first to report cases of concurrent TBE and active *Anaplasma phagocytophilum* infections diagnosed in dogs. Three dogs (3.4% (3/88); CI 95% 0.7–9.6)) were found to be TBEV PCR-positive and concurrently diagnosed with *A. phagocytophilum* infection. However, there are previously reported clinical TBE cases in dogs with high antibody levels against *A. phagocytophilum* and *Ehrlichia canis* in blood serum [32,33]. Concurrent TBE and *Babesia* spp. infections in dogs were reported previously in a case report [15]. Nonetheless, the results of our study indicate that coinfection of tick-borne pathogens in dogs might be underdiagnosed, especially in countries endemic to several tick-borne pathogens.

Isolation of TBEV from tick suspensions in cell cultures significantly increased the number of PCR-positive samples after two serial passages. The results indicate that viral loads in ticks might be too low to be successfully detected using PCR. The prevalence of TBEV RNA in questing ticks is generally less than 1%, although, in some regions in Europe, it has been found to be 2.6–10.8% [34,35,36]. While Belova’s et al. [37] experiment suggests that viral load increases when ticks are feeding on blood, our study found no statistically significant associations between TBE virus presence in dogs and ticks, collected from respective dogs. One possible explanation is that the dogs were infected by ticks that were not collected at the time of the study. Moreover, the period of time of the blood meal or the viral load in the ticks could have been insufficient to infect some of the dogs [38]. A study in neighboring Latvia revealed a high TBEV prevalence in questing *I. ricinus* ticks which varied between 1.7% and 26.6% in 2001 and 1995, respectively, and the prevalence in ticks removed from humans was even higher, reaching 13.2% and 40.9% in 1997 and 1999, respectively [39]. In contrast, a study conducted by Korenberg et al. [40] revealed a higher TBEV prevalence in questing ticks (10.9–38.7%) compared to ticks removed from humans (5.5–11.0%). However, they used different methods for the detection of TBEV: In questing ticks, TBEV was detected after isolation in PK cells and confirmed by direct immunofluorescence test (and additional tests), and TBEV in ticks removed from humans was detected by indirect immunofluorescence assay [40]. Our study shows that virus isolation in cell culture is a more effective method to increase viral load in ticks to detectable levels compared to virus replication in the ticks’ salivary glands during a blood meal. The main reason might be an insufficient length of time for the tick blood meal because most people remove ticks promptly (in around 90% of cases), and reactivation, which is required for some microorganisms before infectivity is attained, does not have a sufficient amount of time to develop [40]. Moreover, other factors, such as host immune response, might have an influence. A study conducted in Sweden analyzed the prevalence of TBEV in detached ticks from human patients and the clinical response of the tick-bitten participants. Two persons developed antibodies against TBEV and one of the patients reported clinical symptoms, despite a lack of TBEV in detached ticks. In addition, one patient did not experience clinical symptoms and did not have antibodies against TBEV, despite the detached tick containing TBEV (1800 copies) [38].

In the present study, the clinical symptoms of PCR-positive dogs were similar to those documented in case reports and other studies of TBE in dogs. The most common symptoms of the first phase of TBE are fever, apathy, and anorexia. Neurological symptoms such as seizures, altered behavior, and consciousness, ataxia/vestibular symptoms, plegia/paresis of one or more limbs, reduced limb muscle tone and spinal reflexes, neck hyperalgesia, cranial nerve deficits, nystagmus, optic neuritis, inability to swallow, and other symptoms were reported in dogs with TBE infection [5,10,32]. Although optic neuritis is a rarely occurring clinical symptom of TBE in dogs, two cases have been previously described [15,32]. In the present study, four dogs could be suspected to have optic neuritis. Three PCR-positive dogs and one dog with high levels of TBE-specific antibodies had transient vision impairment which improved after treatment with corticosteroids. Thus, in endemic areas, TBEV should be considered a possible cause of vision impairment in dogs. Neurological symptoms were present in 31.8% (28/88; CI 95% 22.3–42.6) of PCR-positive dogs. In accordance with our findings, a study conducted in the endemic region of the Czech Republic found that 35% (7/20) of PCR-positive dogs had neurological clinical symptoms [20]. Gastrointestinal symptoms such as vomiting and diarrhea were observed in 37.5% (33/88; CI 95% 27.4–48.5) of PCR-positive dogs. Similar prevalence rates of gastrointestinal symptoms were reported in human TBE patients. A study conducted by Zambito Marsala et al. [41] reported vomiting in 30% (27/89) of cases and nausea in 24.7% (22/89) of cases in human patients. Mickienė et al. [42] reported gastrointestinal complaints in 21.3% (20/94) of TBE patients. The absence of hyperthermia in dogs with seizures was related to a worse outcome. Similarly, a retrospective study of TBE in dogs concluded that hyperthermia was related to a better outcome of disease [5].

## 5. Conclusions

The revealed results of the presence of neurological symptoms in 31.8% (28/88; CI 95% 22.3–42.6) and a death rate of 18.2% (16/88; CI 95% 10.8-27.8) of PCR-positive dogs implies that TBE is a clinically important disease in dogs in such endemic countries as Lithuania. However, TBE is underdiagnosed in dogs because of a lack of awareness about the disease even in TBE-endemic areas. The present study shows that symptomatic treatment and intensive care might be necessary in severe cases and the treatment is more effective in young dogs without concurrent diseases. The timely diagnosis of TBE might help to apply more effective treatment and supportive care strategies and save at least some of the dogs from euthanasia. TBE is therefore recommended to be included in the differential diagnosis of CNS diseases in dogs in endemic areas.

## Figures and Tables

**Figure 1 viruses-15-02265-f001:**
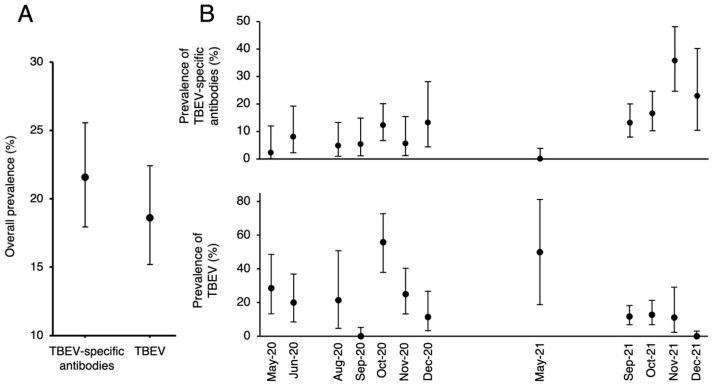
(**A**) The prevalence rate (%) of TBEV RNA and TBEV-specific antibodies in blood/blood serum samples of dogs and (**B**) prevalence rate (%) in different months of sample collection.

**Figure 2 viruses-15-02265-f002:**
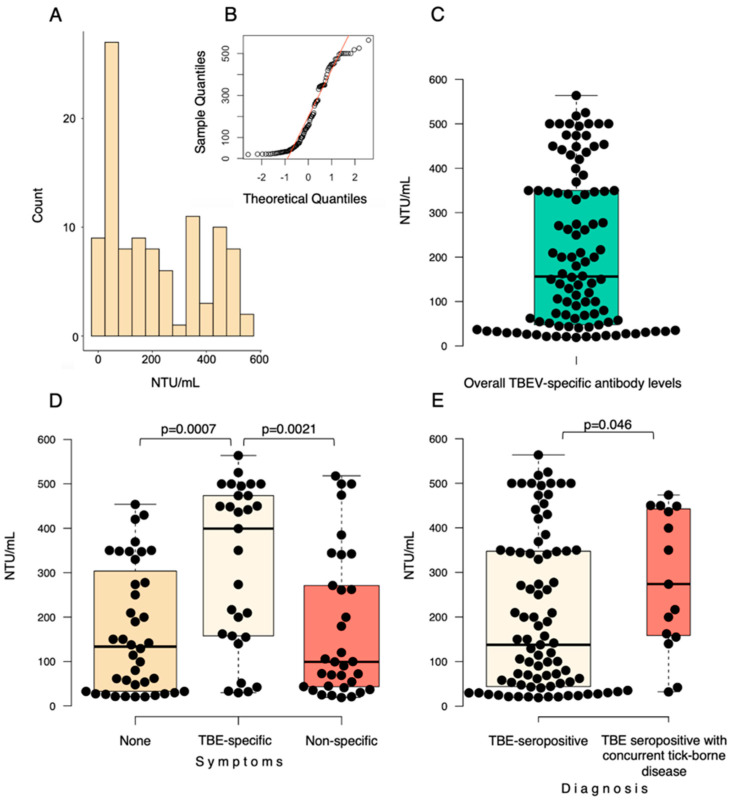
Prevalence of TBEV−specific antibodies. (**A**) Number of seropositive dogs and levels of detected TBEV−specific antibodies. (**B**) Q−Q plot. (**C**) Differences in levels of TBEV−specific antibodies in seropositive dogs. (**D**) Differences in levels of TBEV−specific antibodies in dogs with potentially TBE−specific symptoms, non−specific symptoms, and dogs with non−TBE−related symptoms. (**E**) Differences in levels of TBEV−specific antibodies in dogs with other tick−borne diseases and dogs without other concurrently diagnosed tick−borne diseases.

**Figure 3 viruses-15-02265-f003:**
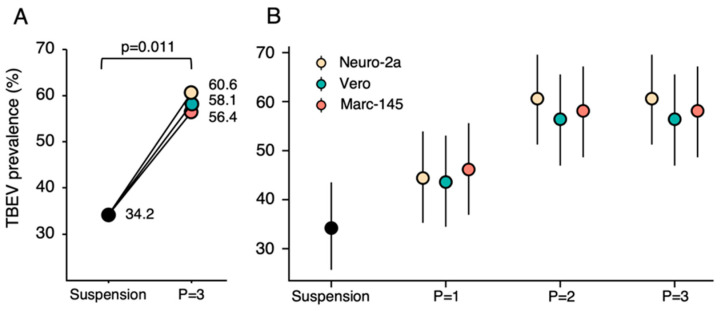
(**A**) Comparison of the prevalence rates (%) of TBEV RNA in tick suspensions and third passage isolates in Neuro-2a, Vero, and Marc-145 cell cultures. (**B**) TBEV RNA prevalence rate (%) changes at each passage in different cell cultures.

## Data Availability

Data will be made available on request.

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
