# Peer review of "The Prevalence, Seroprevalence, and Risk Factors of Tick-Borne Encephalitis Virus in Dogs in Lithuania, a Highly Endemic State"

_viruses, 2023, doi:10.3390/v15112265_

Round 1

Reviewer 1 Report

Comments and Suggestions for Authors

The abstract concisely summarises the research aims and major results, while the introduction presents the topic and its importance. The findings section clarifies Lithuanian dog tick-borne encephalitis virus prevalence and seroprevalence. The detailed examination of viral transmission risk variables is noteworthy. Additional information about the practical relevance and significance of this study in the conclusion and discussion is required. The research might be strengthened by explaining how the results relate to tick-borne encephalitis virus in dogs and possible treatments or preventative strategies.To support the study's assertions and provide future readers context, the references section might use more entries. The study would be more credible and strong with additional relevant literature.The research sheds light on Lithuanian dogs' tick-borne encephalitis virus frequency and risk factors. The abstract, introduction, and findings are good, however, a more detailed conclusion and discussion and a longer list of references will greatly boost the article's impact and contribution.

Author Response

Dear Reviewer,

Thank you for your revision.

“The abstract concisely summarises the research aims and major results, while the introduction presents the topic and its importance. The findings section clarifies Lithuanian dog tick-borne encephalitis virus prevalence and seroprevalence. The detailed examination of viral transmission risk variables is noteworthy. Additional information about the practical relevance and significance of this study in the conclusion and discussion is required. The research might be strengthened by explaining how the results relate to tick-borne encephalitis virus in dogs and possible treatments or preventative strategies.To support the study's assertions and provide future readers context, the references section might use more entries. The study would be more credible and strong with additional relevant literature.The research sheds light on Lithuanian dogs' tick-borne encephalitis virus frequency and risk factors. The abstract, introduction, and findings are good, however, a more detailed conclusion and discussion and a longer list of references will greatly boost the article's impact and contribution.”

Additional information about the practical significance of this study was added to the conclusions. Additional relevant literature was cited in the text and added to the reference list.  

P.11, L.396-401 “However, TBE is underdiagnosed in dogs because of a lack of awareness about the disease even in TBE-endemic areas. The present study shows that symptomatic treatment and intensive care might be necessary in severe cases and the treatment is more effective in young dogs without concurrent diseases. The timely diagnosis of TBE might help to apply more effective treatment and supportive care strategies and save from euthanasia at least some of the dogs.” was added in the conclusions.

The preventative strategies are basic, therefore we did not mention it. The use of acaricides and more frequent grooming with the goal of finding and removing ticks are known preventative strategies of TBEV. The TBE vaccine is currently available only for humans.

On behalf of all Authors,

Evelina Simkute

Reviewer 2 Report

Comments and Suggestions for Authors

In this manuscript titled “The Prevalence, Seroprevalence, and Risk Factors of Tick-borne Encephalitis Virus in Dogs in Lithuania, a Highly Endemic State”, a study was conducted. The manuscript is well-organized and has certain significance. It was addressed a specific gap in the field.However, there are still some problems in this manuscript that need to be revised;

1 The language needs considerable attention.

2 The author can add some model pictures to better display.

3 The reference writing is not standardized

Author Response

Dear Reviewer,

Thank you for your revision.

“In this manuscript titled “The Prevalence, Seroprevalence, and Risk Factors of Tick-borne Encephalitis Virus in Dogs in Lithuania, a Highly Endemic State”, a study was conducted. The manuscript is well-organized and has certain significance. It was addressed a specific gap in the field. However, there are still some problems in this manuscript that need to be revised;“

1 The language needs considerable attention.

The language was revised.

2 The author can add some model pictures to better display.

The authors unanimously agreed that the graphical visualizations already show the results that could be graphically represented. We feel there is no new information that could be supplied that would not duplicate existing visualizations. Furthermore, none of the other reviewers commented on the visualizations, therefore we believe the study's findings are adequately visualized.

 3 The reference writing is not standardized

Corrected.

On behalf of all Authors,

Evelina Simkute

Reviewer 3 Report

Comments and Suggestions for Authors

The study by Evelina Simkute et al. is relevant since the prevalence of tich-borne encephalitic virus in dogs from Lithuania was examined for the first time. It should be pointed out that in Lithuania human tich-borne encephalitic rates is one of the highest in Europe. The identification of virus was based mainly on ELISA and RT-nPCR. Bellow are remarks, authors should more broadly describe statistical analysis.

11     Please equalise the number of decimal places, when you are calculating p vales

22    Titles of subsections (for example, 2.1, 2.2, 3.1 etc.) must be written using the style when all nouns, adjectives and verbs are capitalized

33     L38-39 it is nor clear for me do you refer here to countries (if so, please list them) or locations in countries?

44    L65 please explain in which cases CSF samples were available, it is only in three cases, so I guess it might be related to the conditions of dogs

55   L70 spp. should be in normal style, not italic. Please correct through manuscript

66     L71 after first mentioning of the species, please abbreviate genus name A. phagocytophilum. Please correct through manuscript

77    L105 what is NCR, please add brief information and explain the abbreviation.

88    L106-110 the description is generalized; other investigators would not be able to replicate what you have done. Please mention in the text, cite and describe for what purposes were used exact R programs.

99   Please change title, “genral results” is really unspecific. It might be Prevalence of TBEV in dogs

110.  L125 “The overall rate of death/euthanasia was 18.2% (CI 95% 10.8-27-8).” But in L20-21 “the death/euthanasia rate was 18.2% (CI 95% 10.8-27-8) in 20 PCR-positive dogs” so there is correct in overall sample, or in PCR-positive.

111.  L131-133 the first citing of supplementary material should be in Methods section, also in methods please shortly describe how many breeds were investigated and maybe what was the most common anamnesis.

112.  L149 why here chi square test values are provided? And in other places only p. Have you used different test (maybe chi square, exact)? This methodological aspect must be clearly described and explained in 2.4.

113.  L155-157 description of statistical methods used must be included in Methods 2.4

114.   L164-179 I cannot find Fig.2B citing in the text

115.  L229-230 what was the prevalence of both tich species in analysed dogs? Please include. Also, it clearly visible that that of Ixodes ricinus species was higher, please include significance p value.

116.  L245-249 In methods it was written that Successful virus isolation was confirmed by partial genome sequencing. Where are these data? are all cases confirmed using sequencing or alternatively by RT-nPCR?

Comments on the Quality of English Language

English is good

Author Response

Dear Reviewer,

Thank you very much for your revision.

"The study by Evelina Simkute et al. is relevant since the prevalence of tich-borne encephalitic virus in dogs from Lithuania was examined for the first time. It should be pointed out that in Lithuania human tich-borne encephalitic rates is one of the highest in Europe. The identification of virus was based mainly on ELISA and RT-nPCR. Bellow are remarks, authors should more broadly describe statistical analysis."

We have pointed out the fact of the high prevalence rate of human cases in Lithuania:

P.1, L.33-34 “For the past decade, Lithuania has recorded one of the highest prevalence rates of human TBE cases in Europe”.

11     Please equalise the number of decimal places, when you are calculating p vales 

Corrected.

22    Titles of subsections (for example, 2.1, 2.2, 3.1 etc.) must be written using the style when all nouns, adjectives and verbs are capitalized

Corrected.

33     L38-39 it is nor clear for me do you refer here to countries (if so, please list them) or locations in countries? 

Locations/areas in countries. In Finland 40% on Åland Islands and 6% on Southwestern archipelago [6]. In Denmark, 30.4% of positive samples in five areas were detected by ELISA: 50% in Bornholm, 50%,  34% in Zaeland, 30%  in Funen, and 21% in Jutland. However, only six samples (4.8%) were NT positive (5 of 6 dogs living in Bornholm) [7]. No data on the area of sample collection was provided, although, in the other study conducted in the Bavaria region, the prevalence of TBEV-specific antibodies was 29.2% [8] (The other study conducted in Bavaria was made by Janitza-Futterer D. Serologische Untersuchungen zur endemischen Situation der Infektion mit dem FSME-Virus in einer südbadischen Pferde-und Hundepopulation [Dissertation]. Munich: Ludwig-Maximilians-Universität München (2003).

44    L65 please explain in which cases CSF samples were available, it is only in three cases, so I guess it might be related to the conditions of dogs 

Yes, the collection of CSF samples was related to the condition and severe neurological symptoms of the dogs. Also, in some cases, the owners of the dogs did not agree to take CSF samples and preferred a more conservative treatment or euthanasia of the dogs with severe neurological symptoms.

P.2, L.65-66 was changed to “Additionally, 3 samples of cerebrospinal fluid (CSF) were collected and analyzed”.

55   L70 spp. should be in normal style, not italic. Please correct through manuscript

Corrected.

66     L71 after first mentioning of the species, please abbreviate genus name A. phagocytophilum. Please correct through manuscript

Corrected.

77 L105 what is NCR, please add brief information and explain the abbreviation. 

NCR – noncoding region.

88    L106-110 the description is generalized; other investigators would not be able to replicate what you have done. Please mention in the text, cite and describe for what purposes were used exact R programs. 

We believe that such a description is adequate because only basic level statistical calculations were employed. No complex models were used, let alone specially tailored. In addition, description provided here is conventional and similar to that of many published papers. It should also be noted that basic calculations are usually not discussed at all in the section of methodology, as they are taken for granted.  In regard to the second part of the question, the R language had no influence on the computations and was just mentioned to show the reader how the visualizations were created. The remaining four reviewers seem to agree with our opinion, as none of them left relevant comments. Therefore, we add only information about post-hoc methodology.

99   Please change title, “genral results” is really unspecific. It might be Prevalence of TBEV in dogs

P.3, L.115 Changed to “Prevalence of TBEV and TBEV-Specific Antibodies in Dogs”.

  1. L125 “The overall rate of death/euthanasia was 18.2% (CI 95% 10.8-27-8).” But in L20-21 “the death/euthanasia rate was 18.2% (CI 95% 10.8-27-8) in 20 PCR-positive dogs” so there is correct in overall sample, or in PCR-positive. 

Corrected to “The rate of death/euthanasia was 18.2% (CI 95% 10.8-27.8) in PCR-positive dogs.”

  1. L131-133 the first citing of supplementary material should be in Methods section, also in methods please shortly describe how many breeds were investigated and maybe what was the most common anamnesis. 

P.2, L.69-71 “The present study included dogs of 45 breeds. The most common anamnesis was apathy, inappetence, and gastrointestinal symptoms (Supplementary Material).” was added.

  1. L149 why here chi square test values are provided? And in other places only p. Have you used different test (maybe chi square, exact)? This methodological aspect must be clearly described and explained in 2.4. 

Chi-square test results are provided because this test was employed in our study. Please note that contrary to your statement, this test was mentioned in section 2.4 in the original version of manuscript.

The other calculations place only p value as there were only two groups (quantitative variable and an independent categorical variable) which means it was simply t test. As stated in a previous comment, we will not, following the accepted procedure, describe common and well-known applications of statistical tools.

  1. L155-157 description of statistical methods used must be included in Methods 2.4

Thank you for the notice. We agree this should be included. Therefore, description was added.

P3., L.111-112 “Standardized residuals were employed as post-hoc analysis after statistically significant chi-square tests.” was added.

  1. L164-179 I cannot find Fig.2B citing in the text 

Added to P.4, L.169.

  1. L229-230 what was the prevalence of both tich species in analysed dogs? Please include. Also, it clearly visible that that of Ixodes ricinus species was higher, please include significance p value. 

Prevalence rates added. P value will not be presented as there are no statistically significant associations and we do not want to affect consistency of the text (only statistically significant results are presented in the text).

  1. L245-249 In methods it was written that Successful virus isolation was confirmed by partial genome sequencing. Where are these data? are all cases confirmed using sequencing or alternatively by RT-nPCR?

The sequencing was made only for internal confirmation of the specificity of detected TBEV RNA, therefore the obtained sequences were not sent to GeneBank. Furthermore, obtained 126 bp partial genome sequences based on NCR fragment of TBEV are practically useless for phylogenetic analysis of circulating virus strains.

On behalf of all Authors,

Evelina Simkute

Reviewer 4 Report

Comments and Suggestions for Authors

Simkute E. et al.: The prevalence, seroprevalence, and risk factors of tick-borne encephalitis virus in dogs in Lithuania, a highly endemic state.

The authors made an interesting survey among dogs in Lithuania which results clearly merit publication. The manuscript is concisely written - it may be that overly concisely at places.. Some important parameters thus remain hidden to the reader: I personally view as particularly insufficient the information on how the dog population is in fact represented. Samples’ selection predetermines the results and should be adequately explained. Instead, the authors limit themselves to a laconic statement that samples were “randomly collected” at two veterinary clinics (p.2, l.61-5). But how the dogs presented at those clinics were randomized is a mystery It makes difference whether a purely stochastic - a ‘throw-a-coin’ mechanism was applied or the selection was quasi-random, as if forcing all races to be represented, or so..

My greatest reservation about this study is that it lacks controls – individuals safe from biting ticks (e.g. zoo animals - wolves, dingoes, experimental beagles, or police dogs or pets confined to metropolitan area, etc.) that could be used as a reference. Without a proper control group, it is impossible to adjust objectively cut-offs (p.5, fig.2), and some conclusions are rather speculative. If such a data exists in an authors’ archive, I encourage them to amend it.

Also, considering the fact that this manuscript has been submitted to the section: Human Virology and Viral Diseases, the authors may consider putting more emphasis on an overlap with human TBE, for example, whether dogs could serve sentinel animals for human diseases e.t.c.

In sum, I recommend a moderate revision.

P.1, l.38: either “the family Flaviviridae” or “the genus Flavivirus”

P.2, l.65-6.  “Additionally, when available samples of cerebrospinal fluid (CSF) were obtained. ” – is this sentence complete?

P.2, l.71: Antibodies to Anaplasma…

P.2, l.78: analysis using ELISA…

P.3, l.114 and throughout: “Viral RNA was detected in 18.6% (CI 95% 15.2-22.4) of dog blood/serum samples”  - , pls, in addition to per-cent, always specify the number and the whole, ->> “Viral RNA was detected in 18.6% (88/473; CI 95% 15.2-22.4) of dog blood/serum samples”

P,7, l.260-1 : Saksida et al. [21]. concluded that RT-PCR is an efficient method for an early diagnosis of TBE while specific antibodies are still undetectable

P.7, l.264-7:TBEV-specific antibodies were detected … in 32.1% of dogs with neurological symptoms. Although the results are consistent with reports from other European countries, a substantially greater seroprevalence of 37.5% was found in horses..” – this statement isn’t correct: note that the difference between the proportions of 32.1%, i.e. 28 out of 88, and 37.5%, i,e. 113 out of 301, isn’t statistically significant ! So, there is no “substantially greater seroprevalence”…

P7, l.275 : the Czech Republic

P.7, l.280-2 „reported by Kleeb et al. [5]. We, on the other hand, analyzed randomly collected samples of dogs, while Kleeb et al. [5] conducted a retrospective study of confirmed TBE cases in dogs with clinical symptoms.“

P.8, l.185-6: „Salat et al. [24] found a statistically significant correlation between age and VNT seropositivity“

Captions of Fig. 1-3: panel’s label precedes,, description follows – not the other way round! : „(A) The prevalenc e rate (%) of TBEV RNA and TBEV-specific antibodies in blood/blood serum samples of dogs and (B) prevalence rates (%) in different months of sample collection“

Author Response

Dear Reviewer,

Thank you very much for your revision.

"The authors made an interesting survey among dogs in Lithuania which results clearly merit publication. The manuscript is concisely written - it may be that overly concisely at places.. Some important parameters thus remain hidden to the reader: I personally view as particularly insufficient the information on how the dog population is in fact represented. Samples’ selection predetermines the results and should be adequately explained. Instead, the authors limit themselves to a laconic statement that samples were “randomly collected” at two veterinary clinics (p.2, l.61-5). But how the dogs presented at those clinics were randomized is a mystery It makes difference whether a purely stochastic - a ‘throw-a-coin’ mechanism was applied or the selection was quasi-random, as if forcing all races to be represented, or so.. "

The dogs included in the present study were presented to veterinary clinics because of various diseases or for annual preventive health evaluation. Blood samples of dogs were taken when it was necessary according to the health state of the dog. The randomization comes from different sample sizes collected at different months of the year which were selected randomly. The selection of dogs does not represent all breeds, but it does represent the most common ones in Lithuania. The number of collected samples represents only a small proportion of the dog population in the second-largest city located in the central part of Lithuania. The number of registered dogs is rapidly increasing due to the recent government law to register pets and the number of 14 266 (in 2021) registered dogs increased to 26 296 (in 2023) registered dogs in a period of three years, therefore the actual population size of dogs is unknown.

"My greatest reservation about this study is that it lacks controls – individuals safe from biting ticks (e.g. zoo animals - wolves, dingoes, experimental beagles, or police dogs or pets confined to metropolitan area, etc.) that could be used as a reference. Without a proper control group, it is impossible to adjust objectively cut-offs (p.5, fig.2), and some conclusions are rather speculative. If such a data exists in an authors’ archive, I encourage them to amend it."

We believe that an epidemiologic study does not necessarily include a control group. Also, it would be tremendously hard to find animals safe from tick bites in Lithuania. In addition, it is a complicated task to take blood samples from zoo animals and the majority of them are living in the zoo located near a known TBE focus in an old oak park in Kaunas. Many other city parks in Lithuania are known as TBE foci.

"Also, considering the fact that this manuscript has been submitted to the section: Human Virology and Viral Diseases, the authors may consider putting more emphasis on an overlap with human TBE, for example, whether dogs could serve sentinel animals for human diseases e.t.c."

P.8, L.324-325 <…> and therefore could be used as indicators to evaluate the risk of tick-borne zoonoses in humans.

P8., L.326-328 Furthermore, dogs are known sentinels for TBE foci detection and were used as TBE indicators in countries previously considered to be TBE-free.

In sum, I recommend a moderate revision.

P.1, l.38: either “the family Flaviviridae” or “the genus Flavivirus”

Corrected.

P.2, l.65-6.  “Additionally, when available samples of cerebrospinal fluid (CSF) were obtained. ” – is this sentence complete?

P.2, L.65-66 was changed to “Additionally, 3 samples of cerebrospinal fluid (CSF) were collected and analysed”.

P.2, l.71: Antibodies to Anaplasma…

Corrected.

P.2, l.78: analysis using ELISA…

Corrected.

P.3, l.114 and throughout: “Viral RNA was detected in 18.6% (CI 95% 15.2-22.4) of dog blood/serum samples”  - , pls, in addition to per-cent, always specify the number and the whole, ->> “Viral RNA was detected in 18.6% (88/473; CI 95% 15.2-22.4) of dog blood/serum samples”

Corrected.

P,7, l.260-1 : Saksida et al. [21]. concluded that RT-PCR is an efficient method for an early diagnosis of TBE while specific antibodies are still undetectable 

Corrected.

P.7, l.264-7: „TBEV-specific antibodies were detected … in 32.1% of dogs with neurological symptoms. Although the results are consistent with reports from other European countries, a substantially greater seroprevalence of 37.5% was found in horses..” – this statement isn’t correct: note that the difference between the proportions of 32.1%, i.e. 28 out of 88, and 37.5%, i,e. 113 out of 301, isn’t statistically significant ! So, there is no “substantially greater seroprevalence”…

In the present study, the overall seroprevalence in dogs was 21.6% (102/473). The study in horses does not include TBE cases with neurological symptoms, therefore we compared the overall seroprevalence of dogs.

P7, l.275 : the Czech Republic

Corrected.

P.7, l.280-2 „reported by Kleeb et al. [5]. We, on the other hand, analyzed randomly collected samples of dogs, while Kleeb et al. [5] conducted a retrospective study of confirmed TBE cases in dogs with clinical symptoms.“

Corrected.

P.8, l.185-6: „Salat et al. [24] found a statistically significant correlation between age and VNT seropositivity“

Corrected.

Captions of Fig. 1-3: panel’s label precedes,, description follows – not the other way round! : „(A) The prevalenc e rate (%) of TBEV RNA and TBEV-specific antibodies in blood/blood serum samples of dogs and (B) prevalence rates (%) in different months of sample collection“

Corrected.

On behalf of all Authors,

Evelina Simkute

Round 2

Reviewer 4 Report

Comments and Suggestions for Authors

Simkute E. et al.: The prevalence, seroprevalence, and risk factors of tick-borne encephalitis virus in dogs in Lithuania, a highly endemic state. v. 2

I acknowledge that the manuscsript has been – at least partially – amended to comply with my previous comments. The most prominent design issue of this study – i.e. omission of controls  - however, remains  unamended.  In their response, the authors express a believe (not confidence!) that a control group is of little significance in this kind of study explaining that „it would be tremendously hard“ to recruit adequate controls in Lithuania.  I left this incompleteness to a final decision of the Editor, and, personally, have no further objection or suggestion.

Author Response

Dear Reviewer,

To the best of our knowledge, the control group must be included in experimental studies. None of the epidemiologic studies that we have cited in the manuscript included the control group, only the experimental ones. Our study was epidemiologic, therefore we believe that the study did not require a control group.

On behalf of all Authors,

Evelina Simkute